# Preparation of High Hardness Transparent Coating with Controllable Refractive Index by Sol-Gel Technology

**Xiaolin Liu** [1,†], **Yansong Wang** [2,†] **and Tao Bai** [1,3,*]

1 College of Mechanical Engineering, Donghua University, Shanghai 201620, China; 15161017868@163.com
2 Shanghai Weixing Optical Co., Ltd., Shanghai 201404, China; yswlens@163.com
3 Engineering Research Center of Advanced Textile Machinery, Ministry of Education, Shanghai 201620, China
* Correspondence: baitao@dhu.edu.cn
† Yansong Wang has equal contribution to the first author.

**Abstract:** $SiO_2/ZrO_2$ and $SiO_2/ZrO_2/Al_2O_3$ composite organic–inorganic coatings were prepared by the sol-gel technology. The structure of the coating was characterized by IR, particle size analyzer, SEM, and AFM, respectively. The results showed that $ZrO_2$, $SiO_2$ (inorganic component), and siloxane had undergone a hydrolytic condensation reaction, and the composite organic–inorganic coatings were formed with -O-M-O- (M is Si, Zr) as molecular skeleton network structure. Adding an appropriate ratio of $ZrO_2$ sol, no agglomeration and phase separation occurred, which could significantly improve the refractive index, hardness, and light transmittance of the coatings. $Al_2O_3$ sol could greatly improve the friction resistance of the composite organic–inorganic coatings, and the Bayer ratio of the composite coatings could reach 7.86. By adjusting the proportioning of composite sol solution, the refractive index of the composite transparent coatings could be controlled from 1.52 to 1.65.

**Keywords:** sol-gel technology; refractive index; $ZrO_2$; $Al_2O_3$; coating

## 1. Introduction

In the field of optical materials, since the 1950s, transparent polymer materials with excellent optical properties had entered the research field of human beings, such as acrylic resin (PMMA, PCHMA), polycarbonate (PC), styrene resin (PS, SAN), polyolefin resin (TPX), diethylene glycol diallyl carbonate (CR-39), etc., [1]. Polymer materials are widely used in optical instruments, optical lenses, and other fields because of their excellent optical properties, light weight, and strong impact resistance [2]. However, the inherent defects such as low hardness, not abrasion resistant and low wear-resistance have greatly restricted the service life and application field of optical resin materials. In the development of hard coatings, the wild usages of inorganic coatings and organic coatings have been limited because of their respective defects, so that composite organic–inorganic coatings have become the focus of research in the past ten years [3–7]. At present, the most effective way to solve the problems caused by the soft quality of the resin material is to coat the surface of the optical resin material with high hardness transparent coatings, to increase the wear resistance and scratch resistance.

The most common preparation method of composite organic–inorganic coating is sol-gel technology. Sol-gel technology is hydrolyzed to inorganic nano phase by precursor, which can form membrane alone or composite with other polymers [8]. The reaction is mild and the two-phase dispersion is even, so the sol-gel technology has a great development prospect [9].

Wilkes [10] used metal alkoxide and triethoxysilane sealing precursors such as iminobispropylamine and glycerin to prepare a variety of thin coating materials with excellent wear

resistance, and then applied them to resin substrates which have good mechanical and optical properties. Watling J D [11] used the polysiloxane coatings prepared by 3-glycidoxypropyltrimethoxysilane in alkaline condition, and coated it on glasses. It was proved by Taber corrosion machine that the wear resistance of the coatings was 50% higher than other wear-resistant coatings on the market at present, and the coating had good toughness and longer storage time of hardening liquid. Gilberts J et al. [12] using polyurethane oligomer resin material as coating forming agent, 3-(trimethoxysilyl) propoxymethacrylate (TMSPM) as coupling agent, prepared UV transparent high hardness composite organic–inorganic coating by sol-gel technology, and applied hybrid coating to the plasma-treated PC substrate, and obtained high transparency and scratch-resistance coating after UV treatment. Wouters MEL et al. [13] prepared UV-resistant and high-hardness PU/$SiO_2$ coatings and added conductive polymer into them. Through characterization, it was proved that the more $SiO_2$ content is added, harder this coating is. At the same time the addition of conductive polymer made this coating have antistatic properties, which could reduce the adhesion of dust. The coatings were applied to the material surface of optical devices with good adhesion. He Tao et al. [14] used tetraethyl orthosilicate (TEOS) and trimethoxy (methyl) silane as raw materials which reacted with a certain amount of hydrochloric acid solution, curing agent and leveling agent, etc. at room temperature for 4 h, and then prepared hard coatings. The coatings were applied on the PMMA substrate, and the prepared coatings which had good adhesion could significantly improve the scratch resistance of the surface of the PMMA substrate.

In this paper, inorganic oxides $ZrO_2$ ($n = 2.0 – 2.2$) [15] and $Al_2O_3$ ($n = 1.760$) [16] with high refractive index were used to improve the refractive index and hardness of coatings. A series of $SiO_2$/$ZrO_2$ composite organic–inorganic coatings were prepared by sol-gel technology to improve the refractive index and hardness of the coatings. In order to improve the scratch resistance of thin coatings, $SiO_2$/$Al_2O_3$ sol was introduced into the system and a series of high hardness and frictional resistance transparent coatings of $SiO_2$/$ZrO_2$/$Al_2O_3$ with excellent optical properties were prepared by physical blending method. The refractive index of the coatings could be adjusted by formula to meet the requirements of different refractive index optical lens substrates.

## 2. Experiment

### 2.1. Reagents

Acidic silica sol (concentration of the compound is 30%): Shanghai Silicon Margin Material Technology Co., Ltd. (Shanghai, China); Nano zirconia sol (concentration of the compound is 30%): Shanghai Yifu Industrial Co., Ltd. (Shanghai, China); Nano alumina sol (concentration of the compound is 30%): Shenzhen Jing Materials Chemical Co., Ltd. (Shenzhen, China); Trimethoxy(methyl)silane (concentration of the compound is 98%), Trimethoxy(propyl)silane (concentration of the compound is 98%): Aladdin reagent; glacial acetic acid (analytical purity), methanol (analytical purity): Shanghai Lingfeng Chemical Reagent Co., Ltd. (Shanghai, China); ethylene glycol ether (analytical purity), aluminum isopropoxide (analytical purity) and 3-glycidyloxypropyltrimethoxysilane (KH-560, analytical purity): Aladdin reagent; sodium chloride: Shanghai Lingfeng Chemical Reagent Co., Ltd. (Shanghai, China).

### 2.2. Preparation Technology

#### 2.2.1. Preparation of $SiO_2$/$ZrO_2$ Sol Composite-Hardened Sol Solution

A certain amount of acid silica sol and $ZrO_2$ sol are measured and stirred evenly in the container according to the component ratio. A quantitative amount of glacial acetic acid is added and stirred evenly with the sol for 10 min. An appropriate amount of anhydrous methanol is added to the premix sol, and the pH of the sol is adjusted to 4–5 and stirred evenly for later use. Take an appropriate amount of trimethoxy (methyl) silane and trimethoxy (propyl) silane and mix them evenly.

In a 35 °C water bath, the mixed siloxane solution is slowly added into the silica sol mixture by drip-adding method. The second solution is formed by stirring rapidly and continuously for 1 h. Appropriate amount of ethylene glycol ether, methanol, and aluminum isopropoxide are added to the second solution in order and stirred for 1.5 h. The $SiO_2/ZrO_2$ sol compound-hardening sol solution is prepared by adding an appropriate amount of KH-560 and TEGO440 (a paint additive from Evonik industry) into the above mixture and stirring it continuously for 1 h. [17] The preparation process is shown in Figure 1.

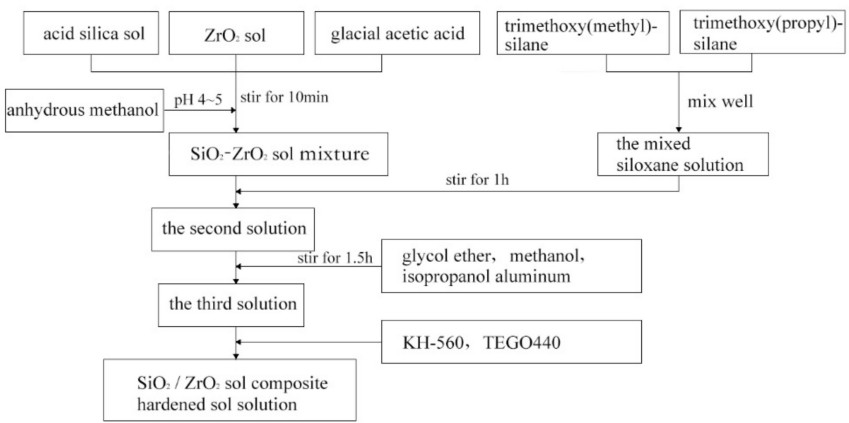

**Figure 1.** Preparation flow chart of $SiO_2/ZrO_2$ sol composite-hardening sol solution.

### 2.2.2. Preparation of $SiO_2/Al_2O_3$ Sol Composite Organic–Inorganic-Hardened Sol Solution

A certain amount of acid silica sol and $Al_2O_3$ sol are measured according to the component ratio and stirred evenly in the container. A quantitative amount of glacial acetic acid is added and stirred evenly with the sol for 10 min. An appropriate amount of anhydric methanol is added to the premix sol, and the pH of the sol is adjusted to 4–5 and stirred evenly for later use. An appropriate amount of propyl trimethoxy (methyl) silane and trimethoxy (propyl) silane were mixed evenly.

For the subsequent steps, refer to Section 2.2.1.

### 2.2.3. Preparation of Controlled Refractive Index Composite-Hardened Sol Based on Aluminum Sol

Select the appropriate amount of A3 sample sol solution in the container according to the ratio, and then take the appropriate amount of Z3 component sol solution. The sol solution is slowly added dropwise to the A3 sample sol solution in a constant temperature water bath at 35 °C, and quickly stirred evenly, physical blending for 4 h.

In Table 1, we list the mixed components of the mixed hardening fluids with different numbers and their mass ratios. ZA1, ZA2, and ZA3 are blended by Z3 and A3 components.

**Table 1.** Proportion and state of compound-hardened sol solution blended with different components.

| Sample | Mixed Components | Mass Ratio | A3 Content% |
|--------|------------------|------------|-------------|
| ZA1 | Z3/A3 | 1/3:1 | 75 |
| ZA2 | Z3/A3 | 1/2:1 | 66.67 |
| ZA3 | Z3/A3 | 1:1 | 50 |

### 2.2.4. Preparation of Transparent Coating with Controlled Refractive Index Based on Aluminum Sol

Cleaning of glass slides: Soak in piranha lotion (30% hydrogen peroxide and 70% concentrated sulfuric acid) for 20 min, ultrasonically clean in deionized water for 30 min, and finally use deionized water to clean 2–3 times, put in an oven to dry for standby. The glass sides dry at 50–70 °C.

Thermal curing of organic-inorganic transparent coating: Apply the prepared composite-hardened sol on the surface of the treated cover glass by pulling, and then put it in an oven and cure at 120 °C for 2 h.

### 2.3. Characterization and Coating Performance Testing

#### 2.3.1. Infrared Spectrum Analysis

Using NEXUS-670 Fourier infrared Raman spectrometer (Waltham, MA, USA), using ATR accessories and potassium bromide tablet to analyze the characteristic bands of the coating before and after curing. The scanning range is 400 cm$^{-1}$–4000 cm$^{-1}$.

#### 2.3.2. Scanning Electron Microscope (SEM) Analysis

The surface morphology of the cured coating is characterized by s-4800 field emission scanning electron microscope (Tokyo, Japanese).

#### 2.3.3. Atomic Force Microscope (AFM) Analysis

The composite-hardened sol solution is thermally cured on the surface of the single-crystal polished silicon wafer. The surface morphology of the composite transparent coating is characterized in the air environment using an atomic force microscope of the IIIa MultiMode model produced by Veeco of the United States (New York, NY, USA).

#### 2.3.4. Particle Size Distribution

The prepared composite-hardened sol solution is diluted to 0.1 g/L with anhydrous methanol as a solvent. The Z average particle size of the composite-hardened sol solution is tested using a Nano ZS nanoparticle size and potential analyzer produced by Malvern, Worcestershire, UK.

#### 2.3.5. Transmittance Analysis

After the surface of the glass slide is thermally cured to form a composite transparent coating, a lambda 950 UV-Vis spectrophotometer produced by Platinum Elmer Instruments (Shanghai, China) Co., Ltd. is used to analyze the transmittance of the sample at a wavelength of 250 nm–800 nm. The coating is prepared by a dip coater with the lifting speed of 1 cm/min.

#### 2.3.6. Hardness Analysis

The national standard GB-T 6739-1996 is adopted to test the pencil hardness of the coatings, and a series of advanced drawing pencils of Zhonghua brand with the hardness of 2B, B, HB, H, 2H, 3H, 4H, 5H, 6H, 7H, 8H, and 9H are used. Place the cover glass slides coated with the transparent wear-resistant coating on the horizontal working table and fix the coating upward. Hold the pencil at about 45°. Push and press on the coating surface so that the pencil lead does not break. The pencil hardness of the coating is denoted as the hardness of the coating when a pencil is scratched on the surface.

#### 2.3.7. Refractive Index Test

Newton defined a new optical constant $n$, which was used to describe the refraction of optical materials [18]. The researchers defined the Newtonian optical constant as the refractive index of the medium—$n$ (refractive index).

$$n = \frac{c}{v} \tag{1}$$

where, $c$ is the propagation speed of light in vacuum.

$v$ is the speed at which light travels in the medium in which it is refracted.

Using a single-sided polished silicon wafer as a substrate, the composite sol is spin-coated on its surface. The m-2000ui J.A.Woollam multifunction intelligent ellipsometric thickness gauge is used to measure the refractive index of the coating. The incident light wavelength is 632.8 nm.

### 2.3.8. Coating Wear Resistance Test

According to the spectacle lens industry standard, the hard-coated lens Bayer experiment is used to determine the wear resistance of the composite organic–inorganic coatings, and the Bayer ratio R is used to characterize the wear resistance of the composite coatings.

The calculation formula of Bayer ratio is shown in Formula (2):

$$\mathrm{R} = \frac{Dstd}{Dtest} \tag{2}$$

In the formula: $Dstd$ is the difference between the haze before and after the standard lens experiment; $Dtest$ is the difference between the haze before and after the experimental lens experiment.

From Formula (2), it can be concluded that the better the wear resistance of the composite coatings is, the smaller the lens fog difference $Dtest$ of the lenses before and after the experiment is, and the larger the Bayer ratio R is. That is to say, the larger the Bayer ratio R is, the better the wear resistance of the composite coatings is.

## 3. Results and Discussion

### 3.1. $SiO_2/ZrO_2$ Sol Composite Organic-Inorganic Transparent Coating

#### 3.1.1. Properties of Composite-Hardening Sol Solution

In this paper, five different samples of $SiO_2/ZrO_2$ sol composite-hardening sols were prepared by adding $ZrO_2$ sol into the $SiO_2$ sol, changing the mass ratio of $SiO_2/ZrO_2$ sol, and five different samples of $SiO_2/ZrO_2$ sol composite organic–inorganic coatings were prepared by thermal curing.

In Table 2, the mass ratio of $SiO_2/ZrO_2$ sol and the content of $ZrO_2$ sol of five components are listed, as well as the state, coating-forming property, and storage time of the composite-hardening sol. Figure 2 shows the state photos of different samples of $SiO_2/ZrO_2$ composite-hardening sol after five days. It can be concluded that when the content of $ZrO_2$ in the component system exceeds 10 wt%, the prepared sol solution gradually changes from the transparent sol to the turbidity emulsion, and the coating prepared after the thermal curing process changes from the transparent to white. Store the compound-hardened sol in a cool place and record the time when the sol precipitates out and delaminates. When the content of $ZrO_2$ is less than 10 wt%, composite-hardening sol solution storage time is more than one hundred and twenty days. When the content of $ZrO_2$ sol increases to 13.33 wt%, composite-hardening sol placed fifteen days after the settling stratification, and the storage time of Z5 composite-hardening sol solution is only three days. As indicated above, when the content of $ZrO_2$ exceeds 10 wt%, the properties of the prepared composite sol solution are not stable.

**Table 2.** $SiO_2/ZrO_2$ sol composite-hardened sol solution with different mass ratio.

| Sample | $SiO_2/ZrO_2$ Sol Mass Ratio | $ZrO_2$ Sol Content wt% | Sol State | COATING-FORMING Property | Storage Time/d |
|--------|------------------------------|-------------------------|-----------|--------------------------|----------------|
| Z1 | 1:1/3 | 5 | Transparency | Well | ≥120 |
| Z2 | 1:1/2 | 6.67 | Transparency | Well | ≥120 |
| Z3 | 1:1 | 10 | Transparency | Well | ≥120 |
| Z4 | 1:2 | 13.33 | Emulsion | Small white | 15 |
| Z5 | 1:3 | 15 | Emulsion | White | 3 |

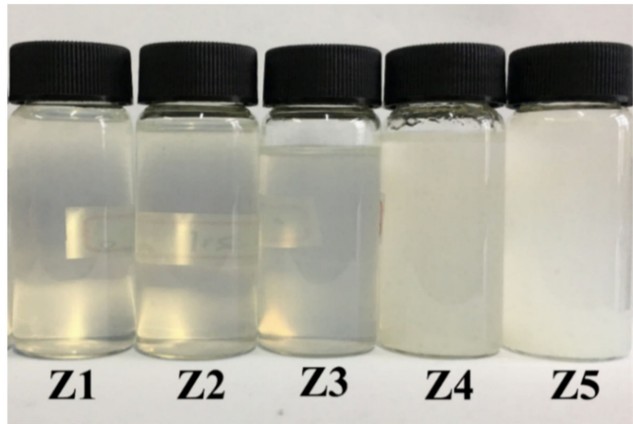

**Figure 2.** SiO$_2$/ZrO$_2$ composite-hardening sol with different proportion.

### 3.1.2. Structural Characterization

Figure 3 is the infrared spectra of SiO$_2$/ZrO$_2$ sol coating with different ratio components. As can be seen from the figure, compared with the infrared spectra of pure ZrO$_2$ sol (Z0) and pure SiO$_2$ sol (Si0), new bonds have been generated. 3420 and 1650 cm$^{-1}$ are the characteristic peaks of -OH stretching vibration and bending vibration. The coatings of different components have an absorption peak here, indicating that unreacted Si-OH and Zr-OH samples still remain in the coatings of different components. 1100 cm$^{-1}$ is the characteristic absorption peak of Si–O–Si, Si–O, and Si–O–C. The coatings of different proportions have a wide absorption peak here, indicating that the hydrolysis condensation of SiO$_2$ sol has occurred in the system. The characteristic absorption peak of Zr-O-Zr is at 470 cm$^{-1}$, indicating that the hydrolysis condensation of ZrO$_2$ sol has occurred in the system. The vibration absorption peak of Zr-O is at 500–850 cm$^{-1}$, which can be seen that there are a few absorption peaks in the composite coatings of different components. The infrared spectrum shows that ZrO$_2$ and SiO$_2$ have undergone a hydrolytic polymerization reaction, forming composite organic–inorganic thin coatings with –O–M–O– (M is Si, Zr) as the molecular skeleton network structure.

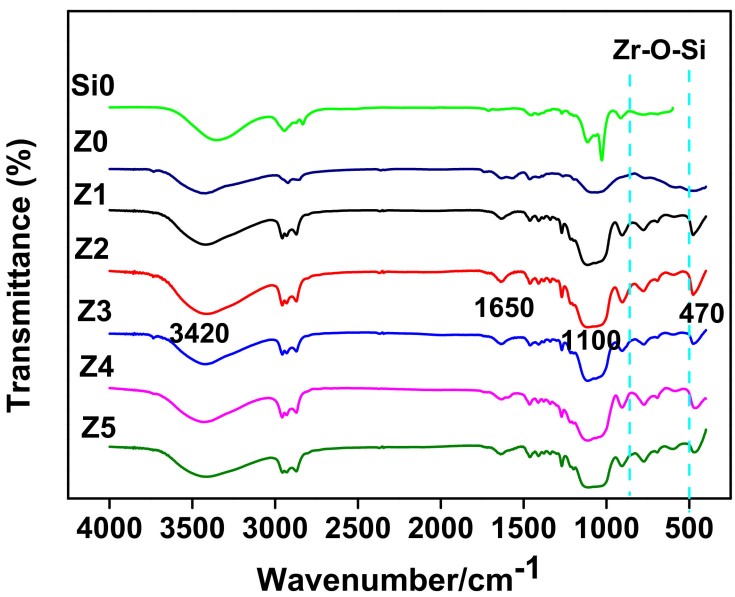

**Figure 3.** Infrared spectrum of Z1, Z2, Z3, Z4, and Z5 composite coatings.

Figure 4 is the ZETA particle size distribution diagram of SiO$_2$/ZrO$_2$ sol with different components. From the figure, it can be seen that the particle size distribution range of Z1, Z2, Z3 composite sol is narrow, and the average particle size is around 100 nm (the smallest particle size is the Z1

which is only 69.98 nm). With the increase of $ZrO_2$ sol content, the particle size distribution of the composite-hardened sol moves to the direction of the increase of particle size, and the distribution range is gradually wider. The particle size distribution range of Z4 composite sol is 19–400 nm and the average particle size is 282 nm. The particle size distribution of Z5 composite sol is divided into two regions, one is 160–600 nm, the other is 4000–5000 nm, indicating that a small number of particles agglomerate in the composite sol-liquid system, which leads to the decrease of the transparency of Z5 composite organic–inorganic coatings and the white surface. It can be concluded that the average particle size of composite-hardened sol increases with the increase of $ZrO_2$ sol content in the system. When the content of $ZrO_2$ sol has increased to 13.33%, the average particle size of the composite sol-liquid system has increased to several hundred nanometers. Agglomeration and phase separation occur easily in the system, which leads to the decline of the stability and storage time of the composite sol-liquid. It also has a great negative impact on the properties of the composite coatings.

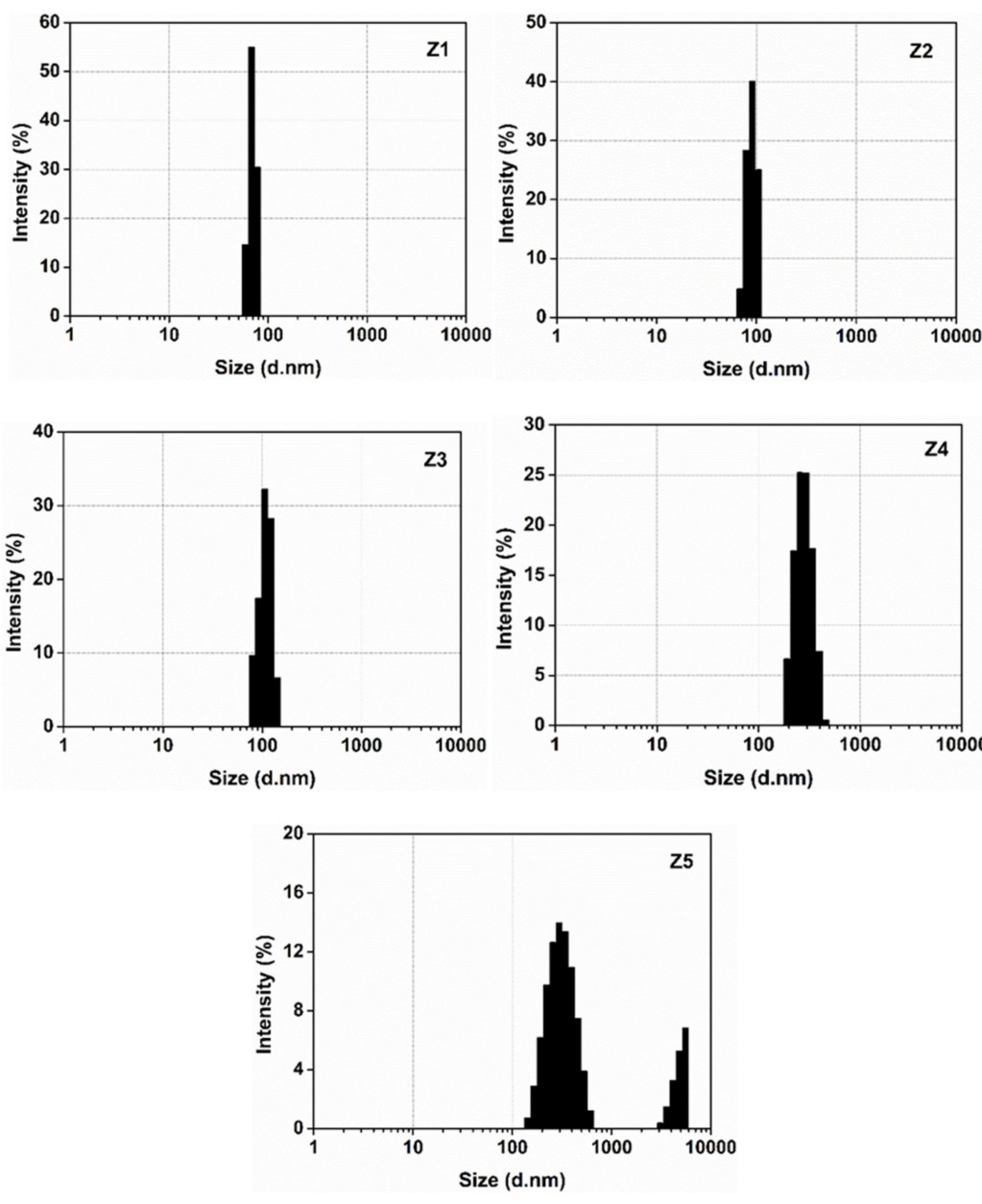

**Figure 4.** Z1, Z2, Z3, Z4, and Z5 composite-hardened sol liquid particle size distribution diagram.

In Table 2, the Z3 coatings with the most stable performance state and the largest $ZrO_2$ content are selected. The Z5 coatings with the worst performance state are selected. The cross section of the composite coating is characterized by field emission scanning electron microscopy (SEM). The results are shown in Figure 5.

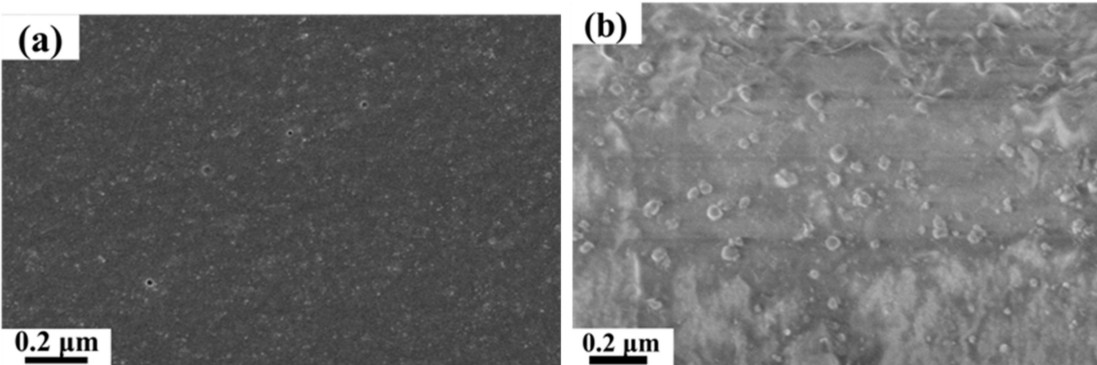

**Figure 5.** (**a**,**b**) SEM images of composite coatings Z3 and Z5.

Figure 5a is the cross-sectional scanning electron microscope image of the Z3 composite coating. It can be observed that the inorganic particles $SiO_2$ and $ZrO_2$ are uniformly distributed in the coating without agglomeration or phase separation. Figure 5b is the cross-sectional scanning electron microscope of Z5 composite organic–inorganic coating. It can be seen that the inorganic particles $SiO_2$ and $ZrO_2$ are not evenly distributed in the coating, and the particle size is not uniform. Agglomeration and phase separation occur in a few places, which greatly affects the optical and mechanical properties of the coatings.

Figure 6 are the atomic force microscope images of $SiO_2/ZrO_2$ sol coatings of Z3 and Z5 components. The number-average roughness (Ra) and RMS roughness (Rq) are 0.570 and 0.734 nm, respectively, indicating that the surface of the composite coating of the Z3 component has good flatness. It can be seen from the figure that there is no obvious phase-difference on the surface. The number-average roughness (Ra) and RMS roughness (Rq) of Z5 composite coating are 1.845 nm and 5.397 nm, respectively, indicating that the surface is relatively rough and the inorganic particles are not completely affected by organic samples. It is embedded, but attached around the skeleton of the network structure. The AFM diagram shows that when the content of $ZrO_2$ in the compound sol solution is 10 wt%, the surface flatness of the coatings is good. When the content of $ZrO_2$ is 15 wt%, the flatness of the composite coating surface is greatly reduced.

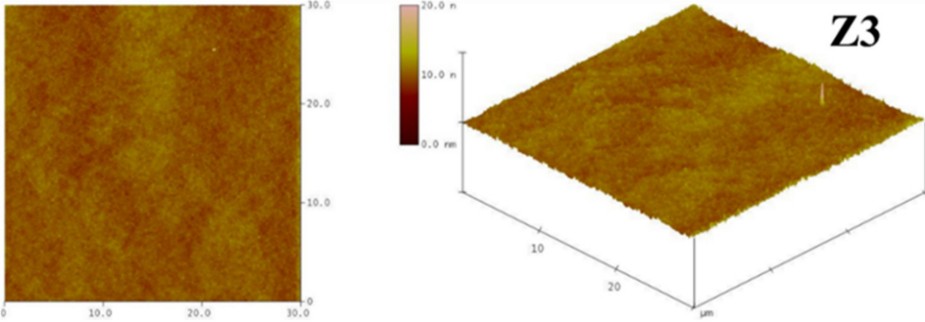

**Figure 6.** *Cont.*

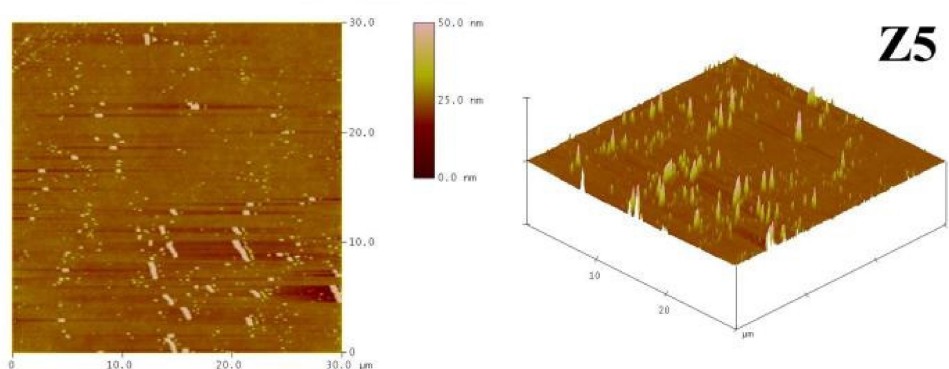

**Figure 6.** AFM diagram of Z3, Z5 composite coating.

### 3.1.3. Performance Test

Figure 7 is the light transmittance curve of Z1, Z2, Z3, Z4, and Z5 composite organic–inorganic transparent coatings. The composite-hardened sol is coated on a glass slide with air. From the figure, it can be drawn that Z1, Z2, Z3, and Z4 coatings have a transmittance of more than 90% in the visible region (400 nm–800 nm), and the highest can reach 94%. With the increase of $ZrO_2$ sol content (Z1, Z2, Z3, Z4, Z5 in turn), the light transmittance of the coatings decreases slightly, but that of Z5 coating decreases obviously. This is because a few inorganic particles agglomerate in the Z5 coating. It greatly affects the light transmittance of the composite coatings. The figure shows that the $SiO_2/ZrO_2$ coatings have good transparency in the visible light region. As the $ZrO_2$ content increases, the light transmittance of the composite coatings decreases. When the $ZrO_2$ content is 15 wt%, the aggregation of inorganic particles can lead to a significant decrease in the light transmittance of the coatings.

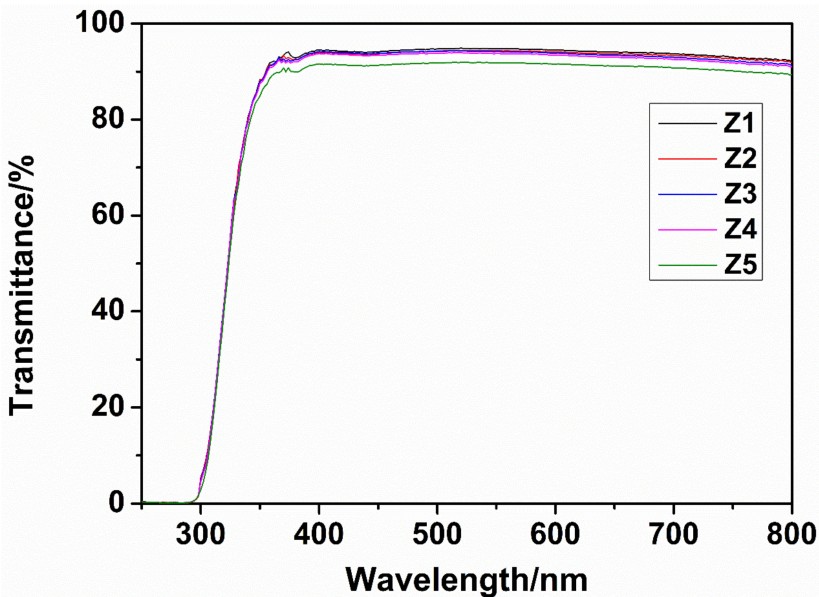

**Figure 7.** Transmittance curves of $SiO_2/ZrO_2$ composite organic–inorganic transparent coatings with different compositions.

As is shown in Figure 8, the refractive index of $SiO_2/ZrO_2$ sol coatings with different components are measured by ellipsometry under the condition that the incident light wavelength is 632.8 nm. With the increase of $ZrO_2$ content, the refractive index of the composite coating increases from 1.592 of Z1 coating to 1.663 of Z5 coating. It shows that $ZrO_2$ sol as an inorganic component has an obvious effect

on the increase of the refractive index of the composite coating. The refractive index of the composite coating can be adjusted by controlling the mass ratio of $ZrO_2/SiO_2$ sol.

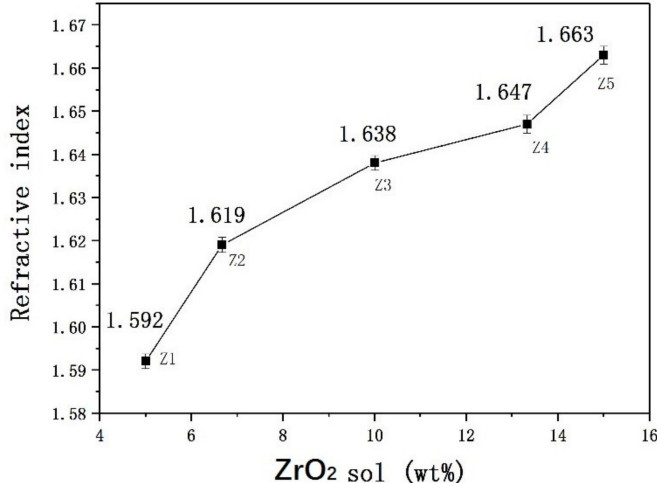

**Figure 8.** Refractive index changes of composite coatings with different $ZrO_2$ sol content.

Table 3 shows the hardness values of $SiO_2/ZrO_2$ sol coatings with different compositions. With the increase of $ZrO_2$ sol content (Z1, Z2, Z3, Z4, Z5 in turn), the hardness of the thin coatings increased. The hardness of Z1 coating is 2H and that of Z4 coating is 6H. As metal nanoparticles, the hardness of $ZrO_2$ is often higher than that of $SiO_2$. With the increase of $ZrO_2$ content, the hardness of the coatings will also increase. Because of the agglomeration of a few inorganic particles in the Z5 composite coating, the stability of the three-dimensional netlike structure of the coating is affected and the hardness of the Z5 composite coating decreases to 5H. The results show that the hardness of coatings is improved by adding $ZrO_2$.

**Table 3.** Hardness of Z1, Z2, Z3, Z4, and Z5 composite coatings.

| Sample | Z1 | Z2 | Z3 | Z4 | Z5 |
|---|---|---|---|---|---|
| Hardness/H | 2 | 3 | 4 | 6 | 5 |

Figure 9 shows the change in Bayer ratio R of the coating of $SiO_2/ZrO_2$ sol composite with different components. It can be seen that the Bayer ratio increases linearly with the increase of $ZrO_2$ sol content in the composite system. The Bayer ratio of Z1 coating is 5.19 and that of Z4 coating is 6.23. When the content of $ZrO_2$ is 15 wt%, agglomeration of inorganic particles occurs in the composite film system. It affects the network structure of the composite film, resulting in a decrease in the wear resistance, and the Bayer ratio is reduced to 5.6. The figure shows that the content of $ZrO_2$ sol in the composite system is in the proper range. The wear resistance of $SiO_2/ZrO_2$ organic–inorganic composite coatings increases with the increase of $ZrO_2$ content which means that the addition of $ZrO_2$ sol has improved the wear resistance of the composite coatings.

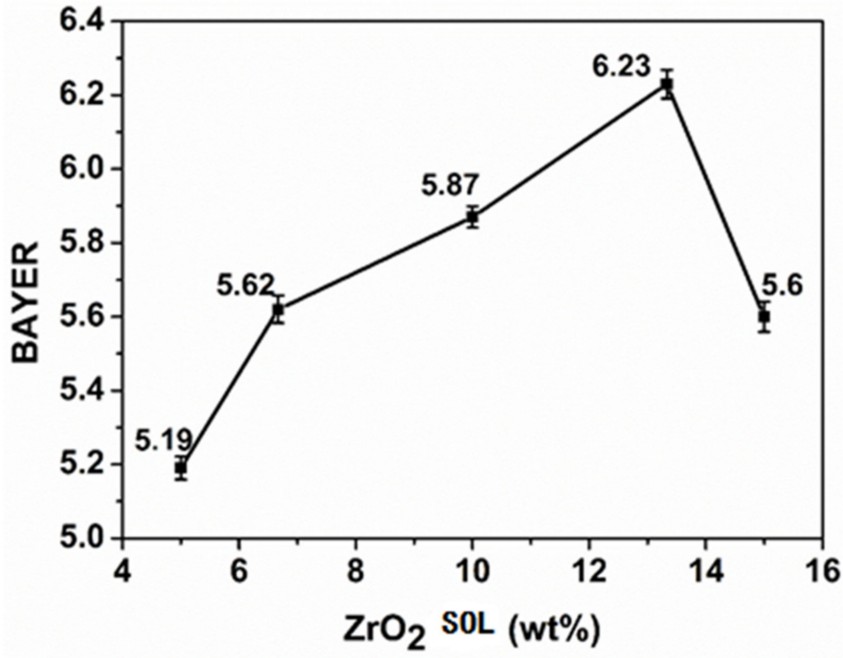

**Figure 9.** Variation of Bayer ratio of composite coatings with different $ZrO_2$ sol content.

### 3.2. $SiO_2/Al_2O_3$ Sol Composite Organic–Inorganic Transparent Coating

In this paper, $Al_2O_3$ sol was added to $SiO_2$ sol-liquid system to change the mass ratio of $SiO_2/Al_2O_3$ sol to prepare different $SiO_2/Al_2O_3$ sol solution samples, and then $SiO_2/Al_2O_3$ composite organic–inorganic coatings were prepared by thermal curing.

The sol state, coating-forming property, and storage time of the sol liquid prepared from different components are shown in Table 4. The composite sol solution of different components was stored in a cool place and the time of precipitation and delamination of the sol was recorded. The composite sol solutions of A1, A2, and A3 are obtained with a long storage time of more than four months, which are very stable. With the increase of the content of $Al_2O_3$ sol, the storage time decreases. For example, the storage time of A4 is two months, and that of A5 is one month. $SiO_2/Al_2O_3$ composite sol has good coating-forming properties and the coatings are transparent and complete.

**Table 4.** $SiO_2/Al_2O_3$ sol solution with different mass ratio.

| Sample | $SiO_2/Al_2O_3$ Sol Mass Ratio | $Al_2O_3$ Sol Content wt% | Sol State | COATING-FORMING Property | Storage Time/d |
|---|---|---|---|---|---|
| A1 | 1:1/3 | 5 | Transparency | Well | ≥120 |
| A2 | 1:1/2 | 6.67 | Transparency | Well | ≥120 |
| A3 | 1:1 | 10 | Transparency | Well | ≥120 |
| A4 | 1:2 | 13.33 | Transparency | Well | ≥60 |
| A5 | 1:3 | 15 | Half-Transparency | Well | 30 |

Figure 10 shows the state photos of different samples of $SiO_2/Al_2O_3$ composite-hardening sol after five days. It can be seen from Figure 10 that A1, A2, A3, and A4 composite sol are transparent, and A5 is slightly half-transparent.

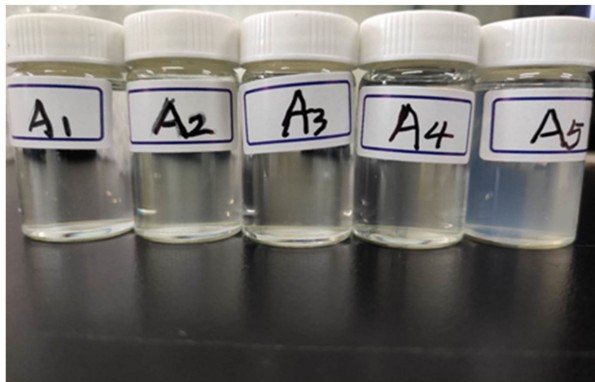

**Figure 10.** $SiO_2/Al_2O_3$ composite-hardening sol with different proportion.

Figure 11 shows the Bayer ratio R changes of $SiO_2/Al_2O_3$ sol composite coatings with different components. It can be observed that with the increase of $Al_2O_3$ sol content in the system, the Bayer ratio of the composite coatings increases linearly, from 6.23 of A1 coating to 7.41 of A5 coating. The result shows that the addition of $Al_2O_3$ sol can greatly improve the wear resistance of the coatings and the wear resistance of the coatings increases with the increase of the content of $Al_2O_3$ sol. According to the storage life of five samples, A3 has the longest storage life and the highest hardness.

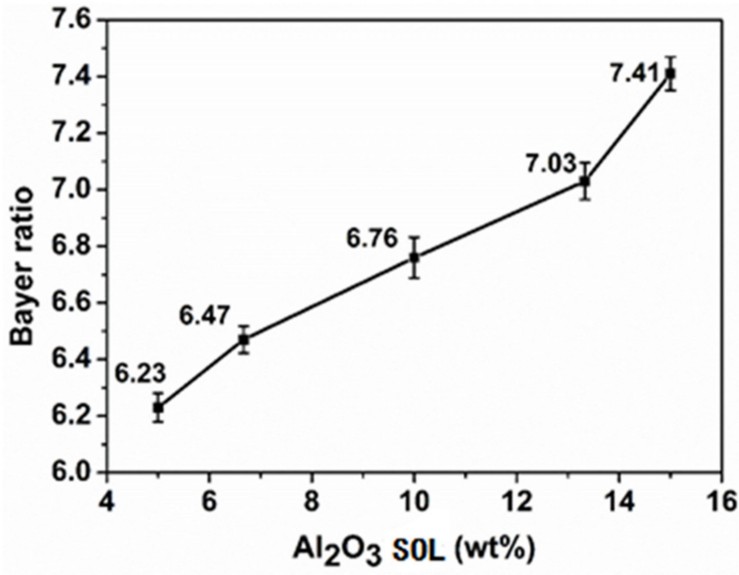

**Figure 11.** Variation of Bayer ratio of composite coatings with different $Al_2O_3$ sol content.

### 3.3. $SiO_2/ZrO_2/Al_2O_3$ Sol Composite Organic–Inorganic Transparent Coating

According to the performance test of the above $SiO_2/ZrO_2$ sol composite organic–inorganic coatings, $ZrO_2$ sol does not significantly improve the wear resistance of the composite coatings. $Al_2O_3$ sol has good adhesion and coating-forming properties which can improve the friction and scratch resistance of the coating surface. Therefore, a series of $SiO_2/Al_2O_3$ composite sol solutions with different compositions are prepared by sol-gel technology and the formulations with excellent properties are selected. Then the optimized $SiO_2/Al_2O_3$ sol solution is added to the high refractive index $SiO_2/ZrO_2$ sol by physical blending method, and different components of $SiO_2/ZrO_2/Al_2O_3$ composite organic–inorganic transparent coatings are prepared.

### 3.3.1. Structural Characterization

Figure 12 shows the particle size distribution of ZA2 composite-hardening sol. It can be seen that the particle size distribution range is small, with an average particle size of 157.93 nm. It indicates that the property of the blended composite-hardening sol is stable.

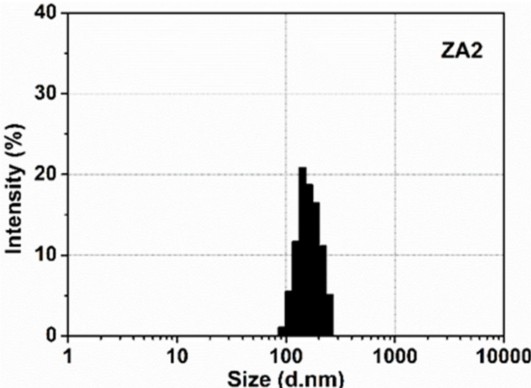

**Figure 12.** The particle size distribution of ZA2 composite-hardened sol.

Figure 13 is a cross-sectional SEM of the coatings of ZA2 component. It can be seen from the figure that inorganic particles ($SiO_2$, $ZrO_2$, $Al_2O_3$) are evenly dispersed in the composite coatings. There is no agglomeration or phase separation. At the same time, there are no obvious cracks on the cross section of the coating, indicating that the composite coatings have good flexibility.

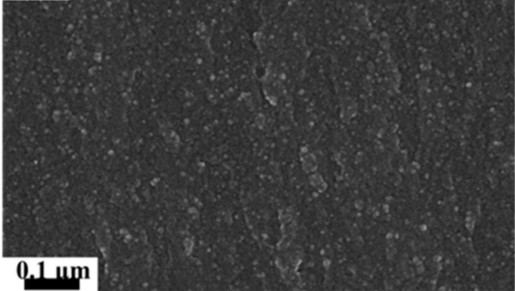

**Figure 13.** SEM of ZA2 composite coatings.

Figure 14 is the AFM pictures of the blend composite coatings of ZA2 component. The number average roughness (Ra) and root mean square roughness (Rq) of ZA2 composite coatings are 0.378 nm and 0.476 nm, respectively. It shows that the coatings have low surface roughness and good flatness.

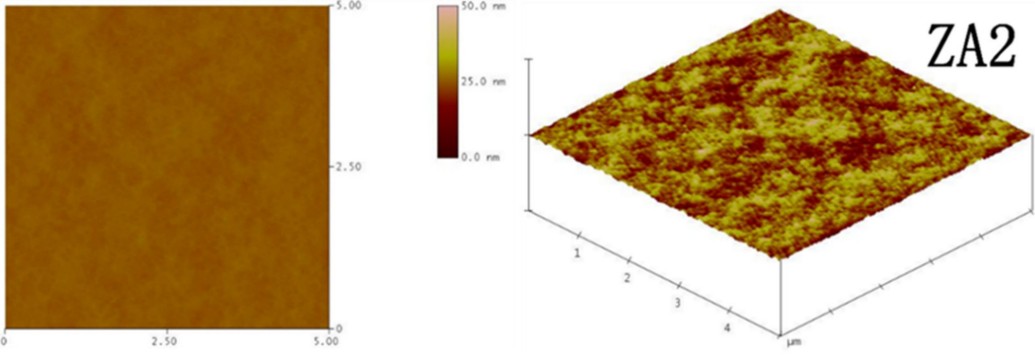

**Figure 14.** AFM diagram of ZA2 composite coating.

3.3.2. Performance Test

Figure 15 shows the light transmittance curve of composite organic–inorganic transparent coatings with different blending components. It can be seen from the figure that in the range of visible light (400–800 nm), the light transmittance of the composite coatings of different components are all above 90% and the maximum reaches 94%. It shows that ZA1 and ZA2 coatings have good transmittance.

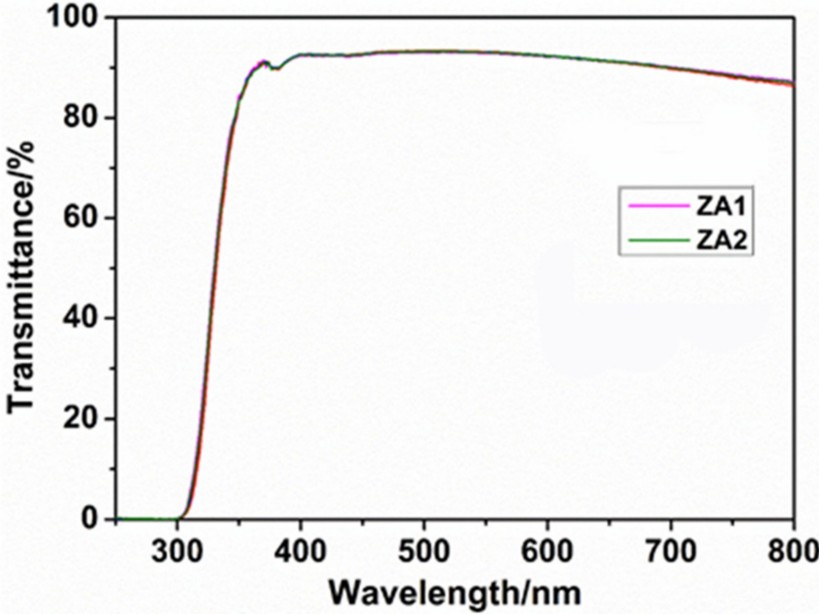

**Figure 15.** Transmittance curve of ZA1 and ZA2 composite coating.

Table 5 lists the refractive index, hardness, and Bayer ratio of composite coatings with different A3 contents. First, it can be observed that by adding Z3 to A3 for physical blending, the refractive index of the prepared coatings is between the values of the two basic hardened sols. The refractive index decreases with the decrease of A3 content, that is, increases with the increase of Z3 content, showing a linear growth relationship. It shows that by controlling the mass ratio of the two basic blended composite-hardened sol solutions, the refractive index of the composite film can be controlled. The blending system is simple in process and convenient in operation. The thickness of the coating is about 8 μm measured by ellipsometer. Second, it can be observed that the hardness values of ZA1 and ZA2 composite coatings are both 5H. It shows that the composite organic–inorganic coatings prepared by physical blending have good hardness. Third, it can be observed that adding A3 to Z3 can improve the Bayer ratio of the composite coatings, and the Bayer ratio increases with the increase of A3 content. The Bayer ratio of Z3 composite coating is 5.87, and that of ZA1 composite coating is 7.86. It shows that the composite coatings have good wear resistance.

**Table 5.** Refractive index, hardness, and Bayer ratio of composite coatings with different A3 content.

| Sample | A3 | ZA1 | ZA2 | Z3 |
|---|---|---|---|---|
| A3 wt/% | 100 | 75 | 66.67 | 0 |
| Refractive index | 1.521 | 1.58 | 1.591 | 1.638 |
| Hardness/H | 4 | 5 | 5 | 4 |
| Bayer Ratio | 6.76 | 7.86 | 7.12 | 5.87 |

*3.4. Mechanism Analysis*

The composite sol solutions of $SiO_2/ZrO_2$ sol and $SiO_2/Al_2O_3$ sol are physically blended. By changing the mass ratio of the two basic composite-hardening sol solutions, the refractive index of

composite organic–inorganic thin coatings can be controlled. As inorganic components, $ZrO_2$ and $SiO_2$ hydrolyze and condense with siloxane, resulting in the formation of composite organic–inorganic coatings with –O–M–O– (M is Si, Zr) as the molecular skeleton network structure [19]. Figure 16a,b shows the cross-linked network structure of –O–Zr–O–Si–O– and –O–Al–O–Zr–O–Si–O– as the molecular skeleton. Propyl group is represented by R and methyl group by R′ in figures. As an inorganic component, the cross-linked network structure of the molecular skeleton can improve the hardness, scratch resistance, and wear resistance of the coatings. The metal ions can improve the optical properties of the coatings, such as refractive index and light transmittance. The alkyl samples around the molecular skeleton, as organic components, provide the coatings with a certain degree of flexibility and improve the adhesion and binding force between the coating and the substrate of the polymer optical material. As metal nanoparticle, $ZrO_2$ is usually much harder than $SiO_2$ particle. With the increase of $ZrO_2$ content, the hardness of composite organic–inorganic transparent coatings will also increase. However, because of the agglomeration of a few inorganic particles in Z5 composite coatings, the stability of the three-dimensional netlike structure of the coatings is affected and the hardness of Z5 composite coatings decreases. As high refractive material ($n = 2.0 - 2.2$), the refractive index of $ZrO_2$ is usually higher than that of $SiO_2$. Moreover, with the increase of $ZrO_2$ content, the refractive index of composite organic–inorganic transparent coatings also increases correspondingly.

(a)　　　　　　　　　　　　　　　(b)

**Figure 16.** The cross-linked network structure of the molecular skeleton, (**a**) –O–Zr–O–Si–O–; (**b**) –O–Al–O–Zr–O–Si–O–.

## 4. Conclusions

In this paper, $SiO_2/ZrO_2$ and $SiO_2/ZrO_2/Al_2O_3$ composite organic–inorganic transparent coatings were prepared by a novel sol-gel technology. The refractive index of the composite organic–inorganic coatings increased linearly with the increase of metal sol content. The coatings refractive index could be controlled in the range of 1.52 to 1.65. By adding $Al_2O_3$ sol, the hardness and friction resistance of the coatings were improved. The hardness and Bayer ratio of the composite coating could reach 5H and 7.86, respectively. The transmittance of the composite organic–inorganic coatings in the visible light region (400–800 nm) was more than 90%, which showed good transmittance. Inorganic particles were evenly dispersed in the composite coatings, and the coatings had low surface roughness and good flatness.

**Author Contributions:** X.L. participated in the analysis of coating preparation mechanism, the writing and modification of the manuscript and the final draft; Y.W. put forward the idea of coating preparation technology for the research content, and wrote the relevant content and final draft of the article; T.B. participated in the coating preparation and performance test process, manuscript writing and modification and final draft; All authors have read and agreed to the published version of the manuscript and take responsibility for the accuracy and authenticity of the research content.

**Funding:** This work was financially supported by "the National Key Research and Development Program of China" (2016YFB0302300).

**Conflicts of Interest:** The authors declare no conflicts of interest.

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
