# Peer review of "Preparation of High Hardness Transparent Coating with Controllable Refractive Index by Sol-Gel Technology"

_coatings, doi:10.3390/coatings10070690_

Round 1

Reviewer 1 Report

Authors, please find the review comments

  1. There is no data related for ZrO2; Al2O3 for choosing this composites. 
  2. ph meter it should be write properly 
  3. The whole 2.1 should be revise, some of the points were confusing. 
  4. How you confirmed the sol reaction is complete after 1.5/1 hour (refer sol preparations), any analytical tool which you used to confirm.?
  5. Figure 1, the spectra is almost similar, how did you characterize?
  6. Figure 6, 7, 8, can you use some statistical model. 
  7. Conclusion should be strengthen

The above suggestion should be necessary to answer or modify the main manuscript prior to publications 

Reviewer 2 Report

The manuscript concerns synthesis and characterization of  SiO2/ZrO2 and SiO2/ZrO2/Al2O3 composite organic-inorganic coatings.

The manuscript is interesting but contains a lot of small, annoying errors that make reading difficult, the text should be improved.

Below is a list of my comments:

- Abstact, line 16; the sol-gel process involves hydrolytic condensation and not polymerization, please correct.

- Introduction, The text is interesting and well written, but reference numbers should be placed in square brackets [ ], not superscript.

- line 43, the term “triethoxysilane-sealed precursors” is unclear, please correct

- lines 46-47 and 50-51, the names of silanes are incorrect.

- line 55: “…them .Through…”; line 57: “…of dust .At the…”; line 58: “…good adhesion .He Tao…” as examples of editing errors, please correct in the whole text.

- line 58; which means "teso", the abbreviation should be in upper case and be explained.

- lines 63-68 and after; “…SiO2/ ZrO2…”; “…SiO2/Al2O3…”; “…SiO2/ ZrO2/Al2O3…”.

please unify.

- Experimental, line 71 and after; what means "specification 30%", whether it relates to the concentration of the compound, please explain.

- lines 73-74; “methyltrimethoxysiloxane and propyltrimethoxysiloxane”, these names are incorrect, please correct.

- line 79 and after (89, 102); ” ph meter”, it should be write pH-meter, please correct.

- lines 117-119, how long the glass plates were washed in piranha solution and which conditions they were dried under? Please add.

- the descriptions of the apparatus and the conditions of measurement are brief, please extend them.

- line 148; the hardness scale should be written without spaces

- line 150; In the case of degrees of angular arc, the degree symbol follows the number without any intervening space, please correct

- Results and discussion: coatings based on SiO2 / ZrO2 have been extensively tested, there is a lack of detailed discussion of the results of other systems, especially ZA.

- line 275; why the refractive index is expressed as a percentage?

- line 375; “…hydrolyzed and polymerized…”, shouldn't there be condensation?

- line 377; “…-Al-Zr-O-Si-…”, the formula is missing an oxygen atom.

- References, references need to be improved and adapted to the requirements of the journal.

- I couldn't find items 1 and 14 from reference list, maybe adding DOI numbers would be helpful.

Author Response

Point 1: Abstact, line 16; the sol-gel process involves hydrolytic condensation and not polymerization, please correct.

Response 1: Thank you very much for your correction. I've changed “polymerization” into “condensation”.

Point 2: Introduction, The text is interesting and well written, but reference numbers should be placed in square brackets [ ], not superscript.

Response 2: Thank you very much for your correction. I've browsed the full text and corrected all such errors.

Point 3: line 43, the term “triethoxysilane-sealed precursors” is unclear, please correct

Response 3: Thank you very much for your correction. I've changed “triethoxysilane-sealed precursors” into “triethoxysilane sealing precursors”

Point 4: lines 46-47 and 50-51, the names of silanes are incorrect.

Response 4: Thank you very much for your correction. I have changed “γ - glycidyl ether oxypropyltrimethoxysilane and methacryl oxy three oxy silane” into “3-Glycidoxypropyltrimethoxysilane and 3-(trimethoxysilyl)propoxymethacrylate (TMSPM)”.

Point 5: line 55: “…them .Through…”; line 57: “…of dust .At the…”; line 58: “…good adhesion .He Tao…” as examples of editing errors, please correct in the whole text.

Response 5: Thank you very much for your correction. I've browsed the full text and corrected all such errors.

Point 6: line 58; which means "teso", the abbreviation should be in upper case and be explained.

Response 6: Thank you very much for your presentation. It's an abbreviation for Tetraethyl orthosilicate.

Point 7: lines 63-68 and after; “…SiO2/ ZrO2…”; “…SiO2/Al2O3…”; “…SiO2/ ZrO2/Al2O3…”.please unify.

Response 7: Thank you very much for your presentation. I've browsed the full text and unify the format.

Point 8: Experimental, line 71 and after; what means "specification 30%", whether it relates to the concentration of the compound, please explain.

Response 8: Thank you very much for your presentation. I've revised some of the errors, such as changing “specification 30%” into “concentration of the compound is 30%” and correcting some spelling errors of the drug name.

Point 9: lines 73-74; “methyltrimethoxysiloxane and propyltrimethoxysiloxane”, these names are incorrect, please correct.

Response 9: Thank you very much for your presentation. I have changed “methyltrimethoxysiloxane and propyltrimethoxysiloxane” into “Trimethoxy(methyl)silane and Trimethoxy(propyl)silane”.

Point 10: line 79 and after (89, 102); ” ph meter”, it should be write pH-meter, please correct.

Response 10: Thank you very much for your correction. I've browsed the full text and corrected all such errors.

Point 11: lines 117-119, how long the glass plates were washed in piranha solution and which conditions they were dried under? Please add.

Response 11:The glass plates were washed for 20 minutes and dried under 50~70℃。

Point 12: the descriptions of the apparatus and the conditions of measurement are brief, please extend them.

Response 12: Thank you very much for your suggestion. Considering that the detailed tests have been described in 3.3, the conventional instruments in ‘2.1. Reagents and instruments’ have been deleted in the revised manuscript.

Point 13: line 148; the hardness scale should be written without spaces

Response 13: Thank you very much for your correction. I've corrected all such errors.

Point 14: line 150; In the case of degrees of angular arc, the degree symbol follows the number without any intervening space, please correct

Response 14: Thank you very much for your correction. I've corrected all such errors.

Point 15: Results and discussion: coatings based on SiO2 / ZrO2 have been extensively tested, there is a lack of detailed discussion of the results of other systems, especially ZA.

Response 15: Thank you very much for your advice. The detailed discussion of ZA was in 3.3 section of the manuscript. Except for SEM, AFM and transmittance research, Refractive index, hardness and Bayer ratio were also compared in Table 5. 

Point 16: line 275; why the refractive index is expressed as a percentage?

Response 16: Thank you very much for your correction. I've corrected all such errors.

Point 17: line 375; “…hydrolyzed and polymerized…”, shouldn't there be condensation?

Response 17: Thank you very much for your correction. I've changed “polymerized” into “condensed”.

Point 18: line 377; “…-Al-Zr-O-Si-…”, the formula is missing an oxygen atom.

Response 18: Thank you very much for your presentation. I've changed “-Al-Zr-O-Si-” into “-Al-Zr-O-Si-O-”.

Point 19: References, references need to be improved and adapted to the requirements of the journal.

Response 19: Thank you very much for your presentation. References have been adapted to the requirements of the journal.

Point 20: I couldn't find items 1 and 14 from reference list, maybe adding DOI numbers would be helpful.

Response 20: Thank you very much for your presentation. References 1 and 14 do not have DOI. I put the Chinese title of the journal below。

[1] 金友. 塑料光学元件赢得重视[J]. 光机电信息, 2004, (02): 22-25.

[14] 何涛, 高长有. 双组分有机硅涂料及其对有机玻璃表面的增强作用[J]. 有机硅材料, 2006, (06): 288-291.

Reviewer 3 Report

Dear authors
The manuscript is not written well.
There are several drawbacks:
1) The introduction is not written well (need more references, better described the novelty, why these concentrations were chosen...)
2) The methods must be described more precisely, a.g. SEM, FTIR
3) The preparation is not described well. Try to draw the flow chart.
3) There is no information about coating thickness, how the particles are spread in the coating (cross-section of the coating), EDS, XPS data.
4) The mechanism of reactions is not presented.
5) There are no figures performed with a digital camera.
More comments are given in the attached file.
There are several typing errors. Please correct them.

Current quality is too low for further evaluation.

Round 2

Reviewer 2 Report

Thank you very much for including my comments in the manuscript.

- there are a few minor errors to correct: lines 100, 112, 122, 125, 290.

- line 127, Table 1, please indicate in the table in mass ratio which oxide is first - in the text this ratio is given inversely (line 112) and it can be misleading

- line 386, the formula is still incorrect, according to figure 16 it should be:

"-Al-O-Zr-O-Si-" or "-O-Al-O-Zr-O-Si-O-", there isn’t Al-Zr bond in the structures.

Author Response

Point 1: There are a few minor errors to correct: lines 100,112,122,125,290.

Response 1: line 100: “SiO2 / ZrO2” has been changed into “SiO2/ZrO2”. line 101: “SiO2 / Al2O3” has been changed into “SiO2/Al2O3”. line 112: “... hardening The sol ...” has been changed into “... hardening. The sol ...”.; “a mass ratio of 1: 1/3 ”has been changed into “a mass ratio of 1:1/3 ”. line 122: “The glass sides dried at 50~70℃” has been changed into “The glass sides dried at 50~70℃”. line 125: “120 °C ” has been changed into “120℃”. line 290: “decreased to 5 H.” has been changed into “decreased to 5H.”

Point 2: line 127, Table 1 please indicate in the table in mass ratio which oxide is first- in the text this ratio is given inversely (line 112) and it can be misleading.

Response 2: Thank you very much for your correction.In the table 1, ZA is represented Z3/A3. To avoid misleading, I use abbreviations instead.

Point 3: line 386, the formula is still incorrect,according to figure 16 it should be: “-Al-O-Zr-O-Si-”or “-O-Al-O-Zr-O-Si-O-”, there isn’t Al-Zr bond in the structures.

Response 3: Thank you very much for your correction. I have corrected this error.

Reviewer 3 Report

Dear authors

The manuscript still needs several improvements. Some of my comments are given as an attachment.

Regards.

Round 3

Reviewer 3 Report

Dear Authors

Thank you for your answers. 

There are still three comments.

  1. Figure 1. Are you sure, that after mixing acid silica sol + ZrO2 + glacial acidic acid you will get silica sol mixture? Correct also the trimethoxy(propyl silane) - typing mistake. 
  2. Figure 3. What is Si0? Add in the figure description. 
  3. Figure 16. The presented cross-linked network structure does not contain the mixed two siloxane solutions.
